# Statistical Inference for Cluster Trees

**Jisu Kim**
Department of Statistics
Carnegie Mellon University
Pittsburgh, USA
jisuk1@andrew.cmu.edu

**Yen-Chi Chen**
Department of Statistics
University of Washington
Seattle, USA
yenchic@uw.edu

**Sivaraman Balakrishnan**
Department of Statistics
Carnegie Mellon University
Pittsburgh, USA
siva@stat.cmu.edu

**Alessandro Rinaldo**
Department of Statistics
Carnegie Mellon University
Pittsburgh, USA
arinaldo@stat.cmu.edu

**Larry Wasserman**
Department of Statistics
Carnegie Mellon University
Pittsburgh, USA
larry@stat.cmu.edu

## Abstract

A cluster tree provides a highly-interpretable summary of a density function by representing the hierarchy of its high-density clusters. It is estimated using the empirical tree, which is the cluster tree constructed from a density estimator. This paper addresses the basic question of quantifying our uncertainty by assessing the statistical significance of topological features of an empirical cluster tree. We first study a variety of metrics that can be used to compare different trees, analyze their properties and assess their suitability for inference. We then propose methods to construct and summarize confidence sets for the unknown true cluster tree. We introduce a partial ordering on cluster trees which we use to prune some of the statistically insignificant features of the empirical tree, yielding interpretable and parsimonious cluster trees. Finally, we illustrate the proposed methods on a variety of synthetic examples and furthermore demonstrate their utility in the analysis of a Graft-versus-Host Disease (GvHD) data set.

## 1 Introduction

Clustering is a central problem in the analysis and exploration of data. It is a broad topic, with several existing distinct formulations, objectives, and methods. Despite the extensive literature on the topic, a common aspect of the clustering methodologies that has hindered its widespread scientific adoption is the dearth of methods for statistical inference in the context of clustering. Methods for inference broadly allow us to quantify our uncertainty, to discern "true" clusters from finite-sample artifacts, as well as to rigorously test hypotheses related to the estimated cluster structure.

In this paper, we study statistical inference for the *cluster tree* of an unknown density. We assume that we observe an i.i.d. sample $\{X_1, \ldots, X_n\}$ from a distribution $\mathbb{P}_0$ with unknown density $p_0$. Here, $X_i \in \mathcal{X} \subset \mathbb{R}^d$. The connected components $\mathcal{C}(\lambda)$, of the upper level set $\{x : p_0(x) \geq \lambda\}$, are called *high-density clusters*. The set of high-density clusters forms a nested hierarchy which is referred to as the *cluster tree*[1] of $p_0$, which we denote as $T_{p_0}$.

Methods for density clustering fall broadly in the space of hierarchical clustering algorithms, and inherit several of their advantages: they allow for extremely general cluster shapes and sizes, and in general do not require the pre-specification of the number of clusters. Furthermore, unlike flat

clustering methods, hierarchical methods are able to provide a multi-resolution summary of the underlying density. The cluster tree, irrespective of the dimensionality of the input random variable, is displayed as a two-dimensional object and this makes it an ideal tool to visualize data. In the context of statistical inference, density clustering has another important advantage over other clustering methods: the object of inference, the cluster tree of the unknown density $p_0$, is clearly specified.

In practice, the cluster tree is estimated from a finite sample, $\{X_1, \ldots, X_n\} \sim p_0$. In a scientific application, we are often most interested in reliably distinguishing topological features genuinely present in the cluster tree of the unknown $p_0$, from topological features that arise due to random fluctuations in the finite sample $\{X_1, \ldots, X_n\}$. In this paper, we focus our inference on the cluster tree of the kernel density estimator, $T_{\widehat{p}_h}$, where $\widehat{p}_h$ is the kernel density estimator,

$$\widehat{p}_h(x) = \frac{1}{nh^d} \sum_{i=1}^{n} K\left(\frac{\|x - X_i\|}{h}\right), \tag{1}$$

where $K$ is a kernel and $h$ is an appropriately chosen bandwidth [2].

To develop methods for statistical inference on cluster trees, we construct a confidence set for $T_{p_0}$, i.e. a collection of trees that will include $T_{p_0}$ with some (pre-specified) probability. A confidence set can be converted to a hypothesis test, and a confidence set shows both statistical and scientific significances while a hypothesis test can only show statistical significances [23, p.155].

To construct and understand the confidence set, we need to solve a few technical and conceptual issues. The first issue is that we need a *metric* on trees, in order to quantify the collection of trees that are in some sense "close enough" to $T_{\widehat{p}_h}$ to be statistically indistinguishable from it. We use the bootstrap to construct tight data-driven confidence sets. However, only some metrics are sufficiently "regular" to be amenable to bootstrap inference, which guides our choice of a suitable metric on trees.

On the basis of a finite sample, the true density is indistinguishable from a density with additional infinitesimal perturbations. This leads to the second technical issue which is that our confidence set invariably contains infinitely complex trees. Inspired by the idea of one-sided inference [9], we propose a partial ordering on the set of all density trees to define simple trees. To find simple representative trees in the confidence set, we prune the empirical cluster tree by removing statistically insignificant features. These pruned trees are valid with statistical guarantees that are simpler than the empirical cluster tree in the proposed partial ordering.

**Our contributions:** We begin by considering a variety of metrics on trees, studying their properties and discussing their suitability for inference. We then propose a method of constructing confidence sets and for visualizing trees in this set. This distinguishes aspects of the estimated tree correspond to real features (those present in the cluster tree $T_{p_0}$) from noise features. Finally, we apply our methods to several simulations, and a Graft-versus-Host Disease (GvHD) data set to demonstrate the usefulness of our techniques and the role of statistical inference in clustering problems.

**Related work:** There is a vast literature on density trees (see for instance the book by Klemelä [16]), and we focus our review on works most closely aligned with our paper. The formal definition of the cluster tree, and notions of consistency in estimation of the cluster tree date back to the work of Hartigan [15]. Hartigan studied the efficacy of single-linkage in estimating the cluster tree and showed that single-linkage is inconsistent when the input dimension $d > 1$. Several fixes to single-linkage have since been proposed (see for instance [21]). The paper of Chaudhuri and Dasgupta [4] provided the first rigorous minimax analysis of the density clustering and provided a computationally tractable, consistent estimator of the cluster tree. The papers [1, 5, 12, 17] propose various modifications and analyses of estimators for the cluster tree. While the question of estimation has been extensively addressed, to our knowledge our paper is the first concerning inference for the cluster tree.

There is a literature on inference for phylogenetic trees (see the papers [13, 10]), but the object of inference and the hypothesized generative models are typically quite different. Finally, in our paper, we also consider various metrics on trees. There are several recent works, in the computational topology literature, that have considered different metrics on trees. The most relevant to our own work, are the papers [2, 18] that propose the functional distortion metric and the interleaving distance on trees. These metrics, however, are NP-hard to compute in general. In Section 3, we consider a variety of computationally tractable metrics and assess their suitability for inference.

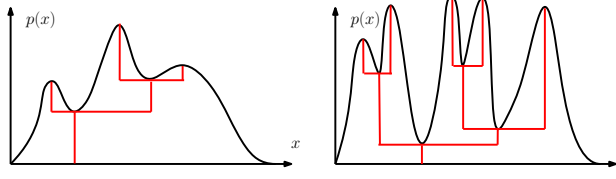

Figure 1: Examples of density trees. Black curves are the original density functions and the red trees are the associated density trees.

## 2 Background and Definitions

We work with densities defined on a subset $\mathcal{X} \subset \mathbb{R}^d$, and denote by $\|.\|$ the Euclidean norm on $\mathcal{X}$. Throughout this paper we restrict our attention to cluster tree estimators that are specified in terms of a function $f : \mathcal{X} \mapsto [0, \infty)$, i.e. we have the following definition:

**Definition 1.** *For any $f : \mathcal{X} \mapsto [0, \infty)$ the* cluster tree *of $f$ is a function $T_f : \mathbb{R} \mapsto 2^{\mathcal{X}}$, where $2^{\mathcal{X}}$ is the set of all subsets of $\mathcal{X}$, and $T_f(\lambda)$ is the set of the connected components of the upper-level set $\{x \in \mathcal{X} : f(x) \geq \lambda\}$. We define the collection of connected components $\{T_f\}$, as $\{T_f\} = \bigcup_{\lambda} T_f(\lambda)$.*

As will be clearer in what follows, working only with cluster trees defined via a function $f$ simplifies our search for metrics on trees, allowing us to use metrics specified in terms of the function $f$. With a slight abuse of notation, we will use $T_f$ to denote also $\{T_f\}$, and write $C \in T_f$ to signify $C \in \{T_f\}$. The cluster tree $T_f$ indeed has a tree structure, since for every pair $C_1, C_2 \in T_f$, either $C_1 \subset C_2$, $C_2 \subset C_1$, or $C_1 \cap C_2 = \emptyset$ holds. See Figure 1 for a graphical illustration of a cluster tree. The formal definition of the tree requires some topological theory; these details are in Appendix B.

In the context of hierarchical clustering, we are often interested in the "height" at which two points or two clusters merge in the clustering. We introduce the merge height from [12, Definition 6]:

**Definition 2.** *For any two points $x, y \in \mathcal{X}$, any $f : \mathcal{X} \mapsto [0, \infty)$, and its tree $T_f$, their **merge height** $m_f(x, y)$ is defined as the largest $\lambda$ such that $x$ and $y$ are in the same density cluster at level $\lambda$, i.e.*

$$m_f(x, y) = \sup \{\lambda \in \mathbb{R} : \text{there exists } C \in T_f(\lambda) \text{ such that } x, y \in C\}.$$

*We refer to the function $m_f : \mathcal{X} \times \mathcal{X} \mapsto \mathbb{R}$ as the merge height function. For any two clusters $C_1, C_2 \in \{T_f\}$, their merge height $m_f(C_1, C_2)$ is defined analogously,*

$$m_f(C_1, C_2) = \sup \{\lambda \in \mathbb{R} : \text{there exists } C \in T_f(\lambda) \text{ such that } C_1, C_2 \subset C\}.$$

One of the contributions of this paper is to construct valid confidence sets for the unknown true tree and to develop methods for visualizing the trees contained in this confidence set. Formally, we assume that we have samples $\{X_1, \ldots, X_n\}$ from a distribution $\mathbb{P}_0$ with density $p_0$.

**Definition 3.** *An asymptotic $(1 - \alpha)$ confidence set, $C_\alpha$, is a collection of trees with the property that*

$$\mathbb{P}_0(T_{p_0} \in C_\alpha) = 1 - \alpha + o(1).$$

We also provide non-asymptotic upper bounds on the $o(1)$ term in the above definition. Additionally, we provide methods to summarize the confidence set above. In order to summarize the confidence set, we define a partial ordering on trees.

**Definition 4.** *For any $f, g : \mathcal{X} \mapsto [0, \infty)$ and their trees $T_f, T_g$, we say $T_f \preceq T_g$ if there exists a map $\Phi : \{T_f\} \to \{T_g\}$ such that for any $C_1, C_2 \in T_f$, we have $C_1 \subset C_2$ if and only if $\Phi(C_1) \subset \Phi(C_2)$.*

With Definition 3 and 4, we describe the confidence set succinctly via some of the *simplest* trees in the confidence set in Section 4. Intuitively, these are trees without statistically insignificant splits.

It is easy to check that the partial order $\preceq$ in Definition 4 is reflexive (i.e. $T_f \preceq T_f$) and transitive (i.e. that $T_{f_1} \preceq T_{f_2}$ and $T_{f_2} \preceq T_{f_3}$ implies $T_{f_1} \preceq T_{f_3}$). However, to argue that $\preceq$ is a partial order, we need to show the antisymmetry, i.e. $T_f \preceq T_g$ and $T_g \preceq T_f$ implies that $T_f$ and $T_g$ are equivalent in some sense. In Appendices A and B, we show an important result: for an appropriate topology on trees, $T_f \preceq T_g$ and $T_g \preceq T_f$ implies that $T_f$ and $T_f$ are *topologically equivalent*.

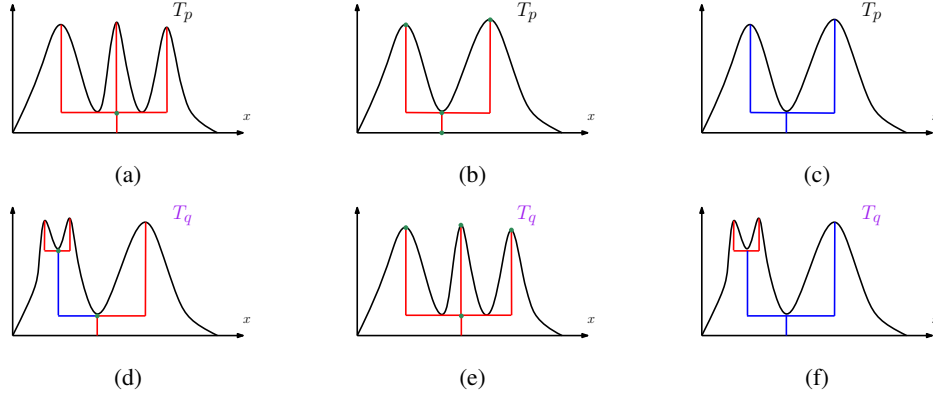

Figure 2: Three illustrations of the partial order $\preceq$ in Definition 4. In each case, in agreement with our intuitive notion of simplicity, the tree on the top ((a), (b), and (c)) is lower than the corresponding tree on the bottom((d), (e), and (f)) in the partial order, i.e. for each example $T_p \preceq T_q$.

The partial order $\preceq$ in Definition 4 matches intuitive notions of the complexity of the tree for several reasons (see Figure 2). Firstly, $T_f \preceq T_g$ implies (number of edges of $T_f$) $\leq$ (number of edges of $T_g$) (compare Figure 2(a) and (d), and see Lemma 6 in Appendix B). Secondly, if $T_g$ is obtained from $T_f$ by adding edges, then $T_f \preceq T_g$ (compare Figure 2(b) and (e), and see Lemma 7 in Appendix B). Finally, the existence of a topology preserving embedding from $\{T_f\}$ to $\{T_g\}$ implies the relationship $T_f \preceq T_g$ (compare Figure 2(c) and (f), and see Lemma 8 in Appendix B).

## 3 Tree Metrics

In this section, we introduce some natural metrics on cluster trees and study some of their properties that determine their suitability for statistical inference. We let $p, q : \mathcal{X} \to [0, \infty)$ be nonnegative functions and let $T_p$ and $T_q$ be the corresponding trees.

### 3.1 Metrics

We consider three metrics on cluster trees, the first is the standard $\ell_\infty$ metric, while the second and third are metrics that appear in the work of Eldridge et al. [12].

$\ell_\infty$ **metric:** The simplest metric is $d_\infty(T_p, T_q) = \|p - q\|_\infty = \sup_{x \in \mathcal{X}} |p(x) - q(x)|$. We will show in what follows that, in the context of statistical inference, this metric has several advantages over other metrics.

**Merge distortion metric:** The merge distortion metric intuitively measures the discrepancy in the merge height functions of two trees in Definition 2. We consider the *merge distortion metric* [12, Definition 11] defined by

$$d_{\mathrm{M}}(T_p, T_q) = \sup_{x,y \in \mathcal{X}} |m_p(x, y) - m_q(x, y)|.$$

The merge distortion metric we consider is a special case of the metric introduced by Eldridge et al. [12][3]. The merge distortion metric was introduced by Eldridge et al. [12] to study the convergence of cluster tree estimators. They establish several interesting properties of the merge distortion metric: in particular, the metric is stable to perturbations in $\ell_\infty$, and further, that convergence in the merge distortion metric strengthens previous notions of convergence of the cluster trees.

**Modified merge distortion metric:** We also consider the *modified merge distortion metric* given by

$$d_{\mathrm{MM}}(T_p, T_q) = \sup_{x,y \in \mathcal{X}} |d_{T_p}(x, y) - d_{T_q}(x, y)|,$$

where $d_{T_p}(x, y) = p(x) + p(y) - 2m_p(x, y)$, which corresponds to the (pseudo)-distance between $x$ and $y$ *along* the tree. The metric $d_{\mathrm{MM}}$ is used in various proofs in the work of Eldridge et al. [12].

It is sensitive to both distortions of the merge heights in Definition 2, as well as of the underlying densities. Since the metric captures the distortion of distances between points along the tree, it is in some sense most closely aligned with the cluster tree. Finally, it is worth noting that unlike the interleaving distance and the functional distortion metric [2, 18], the three metrics we consider in this paper are quite simple to approximate to a high-precision.

## 3.2 Properties of the Metrics

The following Lemma gives some basic relationships between the three metrics $d_\infty$, $d_M$ and $d_{MM}$. We define $p_{\inf} = \inf_{x \in \mathcal{X}} p(x)$, and $q_{\inf}$ analogously, and $a = \inf_{x \in \mathcal{X}} \{p(x) + q(x)\} - 2 \min\{p_{\inf}, q_{\inf}\}$. Note that when the Lebesgue measure $\mu(\mathcal{X})$ is infinite, then $p_{\inf} = q_{\inf} = a = 0$.

**Lemma 1.** *For any densities $p$ and $q$, the following relationships hold: (i) When $p$ and $q$ are continuous, then $d_\infty(T_p, T_q) = d_M(T_p, T_q)$. (ii) $d_{MM}(T_p, T_q) \leq 4 d_\infty(T_p, T_q)$. (iii) $d_{MM}(T_p, T_q) \geq d_\infty(T_p, T_q) - a$, where $a$ is defined as above. Additionally when $\mu(\mathcal{X}) = \infty$, then $d_{MM}(T_p, T_q) \geq d_\infty(T_p, T_q)$.*

The proof is in Appendix F. From Lemma 1, we can see that under a mild assumption (continuity of the densities), $d_\infty$ and $d_M$ are equivalent. We note again that the work of Eldridge et al. [12] actually defines a family of merge distortion metrics, while we restrict our attention to a canonical one. We can also see from Lemma 1 that while the modified merge metric is not equivalent to $d_\infty$, it is usually multiplicatively sandwiched by $d_\infty$.

Our next line of investigation is aimed at assessing the suitability of the three metrics for the task of statistical inference. Given the strong equivalence of $d_\infty$ and $d_M$ we focus our attention on $d_\infty$ and $d_{MM}$. Based on prior work (see [7, 8]), the large sample behavior of $d_\infty$ is well understood. In particular, $d_\infty(T_{\widehat{p}_h}, T_{p_0})$ converges to the supremum of an appropriate Gaussian process, on the basis of which we can construct confidence intervals for the $d_\infty$ metric.

The situation for the metric $d_{MM}$ is substantially more subtle. One of our eventual goals is to use the non-parametric bootstrap to construct valid estimates of the confidence set. In general, a way to assess the amenability of a functional to the bootstrap is via *Hadamard differentiability* [24]. Roughly speaking, Hadamard-differentiability is a type of *statistical stability*, that ensures that the functional under consideration is stable to perturbations in the input distribution. In Appendix C, we formally define Hadamard differentiability and prove that $d_{MM}$ is *not* point-wise Hadamard differentiable. This does not completely rule out the possibility of finding a way to construct confidence sets based on $d_{MM}$, but doing so would be difficult and so far we know of no way to do it.

In summary, based on computational considerations we eliminate the interleaving distance and the functional distortion metric [2, 18], we eliminate the $d_{MM}$ metric based on its unsuitability for statistical inference and focus the rest of our paper on the $d_\infty$ (or equivalently $d_M$) metric which is both computationally tractable and has well understood statistical behavior.

## 4 Confidence Sets

In this section, we consider the construction of valid confidence intervals centered around the kernel density estimator, defined in Equation (1). We first observe that a fixed bandwidth for the KDE gives a dimension-free rate of convergence for estimating a cluster tree. For estimating a density in high dimensions, the KDE has a poor rate of convergence, due to a decreasing bandwidth for simultaneously optimizing the bias and the variance of the KDE.

When estimating a cluster tree, the bias of the KDE does not affect its cluster tree. Intuitively, the cluster tree is a shape characteristic of a function, which is not affected by the bias. Defining the *biased* density, $p_h(x) = \mathbb{E}[\widehat{p}_h(x)]$, two cluster trees from $p_h$ and the true density $p_0$ are equivalent with respect to the topology in Appendix A, if $h$ is small enough and $p_0$ is regular enough:

**Lemma 2.** *Suppose that the true unknown density $p_0$, has no non-degenerate critical points [4], then there exists a constant $h_0 > 0$ such that for all $0 < h \leq h_0$, the two cluster trees, $T_{p_0}$ and $T_{p_h}$ have the same topology in Appendix A.*

From Lemma 2, proved in Appendix G, a fixed bandwidth for the KDE can be applied to give a dimension-free rate of convergence for estimating the cluster tree. Instead of decreasing bandwidth $h$ and inferring the cluster tree of the true density $T_{p_0}$ at rate $O_P(n^{-2/(4+d)})$, Lemma 2 implies that we can fix $h > 0$ and infer the cluster tree of the biased density $T_{p_h}$ at rate $O_P(n^{-1/2})$ *independently of the dimension*. Hence a fixed bandwidth crucially enhances the convergence rate of the proposed methods in high-dimensional settings.

## 4.1 A data-driven confidence set

We recall that we base our inference on the $d_\infty$ metric, and we recall the definition of a valid confidence set (see Definition 3). As a conceptual first step, suppose that for a specified value $\alpha$ we could compute the $1 - \alpha$ quantile of the distribution of $d_\infty(T_{\widehat{p}_h}, T_{p_h})$, and denote this value $t_\alpha$. Then a valid confidence set for the unknown $T_{p_h}$ is $C_\alpha = \{T : d_\infty(T, T_{\widehat{p}_h}) \le t_\alpha\}$. To estimate $t_\alpha$, we use the bootstrap. Specifically, we generate $B$ bootstrap samples, $\{\widetilde{X}_1^1, \cdots, \widetilde{X}_n^1\}, \ldots, \{\widetilde{X}_1^B, \cdots, \widetilde{X}_n^B\}$, by sampling with replacement from the original sample. On each bootstrap sample, we compute the KDE, and the associated cluster tree. We denote the cluster trees $\{\widetilde{T}_{p_h}^1, \ldots, \widetilde{T}_{p_h}^B\}$. Finally, we estimate $t_\alpha$ by

$$\widehat{t}_\alpha = \widehat{F}^{-1}(1 - \alpha), \quad \text{where} \quad \widehat{F}(s) = \frac{1}{B} \sum_{i=1}^n \mathbb{I}(d_\infty(\widetilde{T}_{p_h}^i, T_{\widehat{p}_h}) < s).$$

Then the data-driven confidence set is $\widehat{C}_\alpha = \{T : d_\infty(T, \widehat{T}_h) \le \widehat{t}_\alpha\}$. Using techniques from [8, 7], the following can be shown (proof omitted):

**Theorem 3.** *Under mild regularity conditions on the kernel[5], we have that the constructed confidence set is asymptotically valid and satisfies,*

$$\mathbb{P}\left(T_h \in \widehat{C}_\alpha\right) = 1 - \alpha + O\left(\left(\frac{\log^7 n}{nh^d}\right)^{1/6}\right).$$

Hence our data-driven confidence set is consistent at dimension independent rate. When $h$ is a fixed small constant, Lemma 2 implies that $T_{p_0}$ and $T_{p_h}$ have the same topology, and Theorem 3 guarantees that the non-parametric bootstrap is consistent at a dimension independent $O(((\log n)^7/n)^{1/6})$ rate. For reasons explained in [8], this rate is believed to be optimal.

## 4.2 Probing the Confidence Set

The confidence set $\widehat{C}_\alpha$ is an infinite set with a complex structure. Infinitesimal perturbations of the density estimate are in our confidence set and so this set contains very complex trees. One way to understand the structure of the confidence set is to focus attention on simple trees in the confidence set. Intuitively, these trees only contain topological features (splits and branches) that are sufficiently strongly supported by the data.

We propose two *pruning* schemes to find trees, that are simpler than the empirical tree $T_{\widehat{p}_h}$ that are in the confidence set. Pruning the empirical tree aids visualization as well as de-noises the empirical tree by eliminating some features that arise solely due to the stochastic variability of the finite-sample. The algorithms are (see Figure 3):
1. **Pruning only leaves:** Remove all leaves of length less than $2\widehat{t}_\alpha$ (Figure 3(b)).
2. **Pruning leaves and internal branches:** In this case, we first prune the leaves as above. This yields a new tree. Now we again prune (using cumulative length) any leaf of length less than $2\widehat{t}_\alpha$. We continue iteratively until all remaining leaves are of cumulative length larger than $2\widehat{t}_\alpha$ (Figure 3(c)).

In Appendix D.2 we formally define the pruning operation and show the following. The remaining tree $\widetilde{T}$ after either of the above pruning operations satisfies: (i) $\widetilde{T} \preceq T_{\widehat{p}_h}$, (ii) there exists a function $f$ whose tree is $\widetilde{T}$, and (iii) $\widetilde{T} \in \widehat{C}_\alpha$ (see Lemma 10 in Appendix D.2). In other words, we identified a valid tree with a statistical guarantee that is simpler than the original estimate $T_{\widehat{p}_h}$. Intuitively, some of the statistically insignificant features have been removed from $T_{\widehat{p}_h}$. We should point out, however,

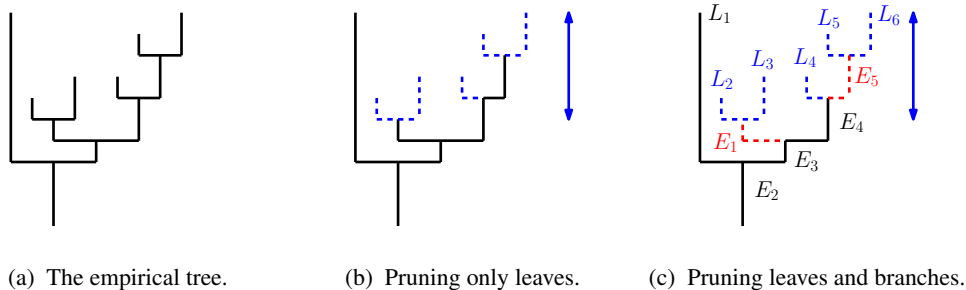

(a) The empirical tree.      (b) Pruning only leaves.      (c) Pruning leaves and branches.

Figure 3: Illustrations of our two pruning strategies. (a) shows the empirical tree. In (b), leaves that are insignificant are pruned, while in (c), insignificant internal branches are further pruned top-down.

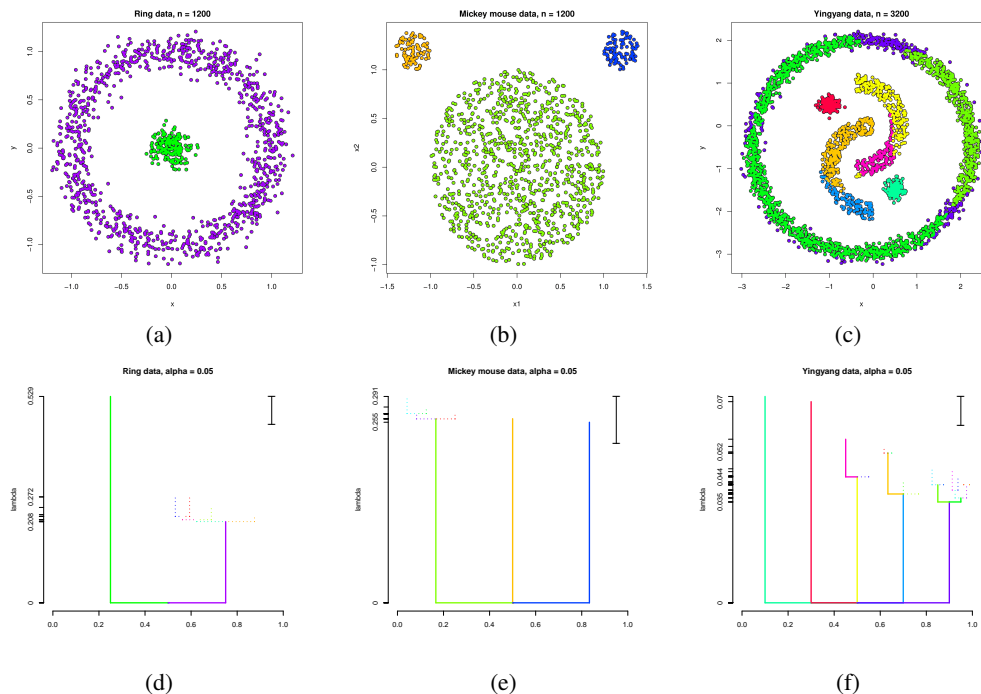

Figure 4: Simulation examples. (a) and (d) are the ring data; (b) and (e) are the mickey mouse data; (c) and (f) are the yingyang data. The solid lines are the pruned trees; the dashed lines are leaves (and edges) removed by the pruning procedure. A bar of length $2\widehat{t}_\alpha$ is at the top right corner. The pruned trees recover the actual structure of connected components.

that there may exist other trees that are simpler than $T_{\widehat{p}_h}$ that are in $\widehat{C}_\alpha$. Ideally, we would like to have an algorithm that identifies all trees in the confidence set that are minimal with respect to the partial order $\preceq$ in Definition 4. This is an open question that we will address in future work.

# 5   Experiments

In this section, we demonstrate the techniques we have developed for inference on synthetic data, as well as on a real dataset.

## 5.1   Simulated data

We consider three simulations: the ring data (Figure 4(a) and (d)), the Mickey Mouse data (Figure 4(b) and (e)), and the yingyang data (Figure 4(c) and (f)). The smoothing bandwidth is chosen by the Silverman reference rule [20] and we pick the significance level $\alpha = 0.05$.

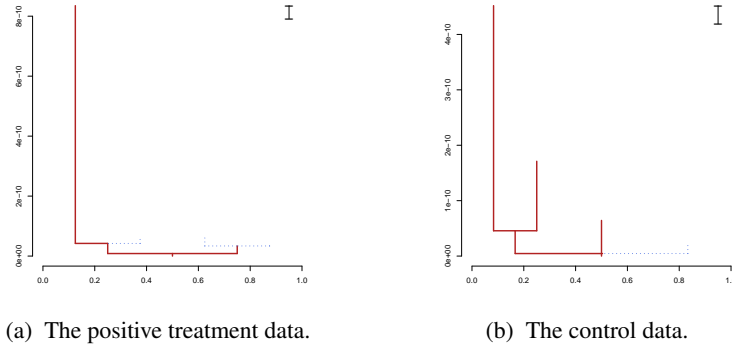

(a) The positive treatment data.       (b) The control data.

Figure 5: The GvHD data. The solid brown lines are the remaining branches after pruning; the blue dashed lines are the pruned leaves (or edges). A bar of length $2\hat{t}_\alpha$ is at the top right corner.

**Example 1: The ring data.** (Figure 4(a) and (d)) The ring data consists of two structures: an outer ring and a center node. The outer circle consists of 1000 points and the central node contains 200 points. To construct the tree, we used $h = 0.202$.

**Example 2: The Mickey Mouse data.** (Figure 4(b) and (e)) The Mickey Mouse data has three components: the top left and right uniform circle (400 points each) and the center circle (1200 points). In this case, we select $h = 0.200$.

**Example 3: The yingyang data.** (Figure 4(c) and (f)) This data has 5 connected components: outer ring (2000 points), the two moon-shape regions (400 points each), and the two nodes (200 points each). We choose $h = 0.385$.

Figure 4 shows those data ((a), (b), and (c)) along with the pruned density trees (solid parts in (d), (e), and (f)). Before pruning the tree (both solid and dashed parts), there are more leaves than the actual number of connected components. But after pruning (only the solid parts), every leaf corresponds to an actual connected component. This demonstrates the power of a good pruning procedure.

## 5.2 GvHD dataset

Now we apply our method to the GvHD (Graft-versus-Host Disease) dataset [3]. GvHD is a complication that may occur when transplanting bone marrow or stem cells from one subject to another [3]. We obtained the GvHD dataset from R package 'mclust'. There are two subsamples: the control sample and the positive (treatment) sample. The control sample consists of 9083 observations and the positive sample contains 6809 observations on 4 biomarker measurements ($d = 4$). By the normal reference rule [20], we pick $h = 39.1$ for the positive sample and $h = 42.2$ for the control sample. We set the significance level $\alpha = 0.05$.

Figure 5 shows the density trees in both samples. The solid brown parts are the remaining components of density trees after pruning and the dashed blue parts are the branches removed by pruning. As can be seen, the pruned density tree of the positive sample (Figure 5(a)) is quite different from the pruned tree of the control sample (Figure 5(b)). The density function of the positive sample has fewer bumps (2 significant leaves) than the control sample (3 significant leaves). By comparing the pruned trees, we can see how the two distributions differ from each other.

## 6  Discussion

There are several open questions that we will address in future work. First, it would be useful to have an algorithm that can find all trees in the confidence set that are minimal with respect to the partial order $\preceq$. These are the simplest trees consistent with the data. Second, we would like to find a way to derive valid confidence sets using the metric $d_{\text{MM}}$ which we view as an appealing metric for tree inference. Finally, we have used the Silverman reference rule [20] for choosing the bandwidth but we would like to find a bandwidth selection method that is more targeted to tree inference.

## Footnotes

[1]It is also referred to as the density tree or the level-set tree.

[2] We address computing the tree $T_{\widehat{p}_h}$, and the choice of bandwidth in more detail in what follows.

[3]They further allow flexibility in taking a sup over a subset of $\mathcal{X}$.

[4]The Hessian of $p_0$ at every critical point is non-degenerate. Such functions are known as Morse functions.

[5]See Appendix D.1 for details.

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
