[Supplementary Material]

# A   Topological Preliminaries

The goal of this section is to define an appropriate topology on the cluster tree $T_f$ in Definition 1. Defining an appropriate topology for the cluster tree $T_f$ is important in this paper for several reasons: (1) the topology gives geometric insight for the cluster tree, (2) homeomorphism (topological equivalence) is connected to equivalence in the partial order $\preceq$ in Definition 4, and (3) the topology gives a justification for using a fixed bandwidth $h$ for constructing confidence set $\widehat{C}_\alpha$ as in Lemma 2 to obtain faster rates of convergence.

We construct the topology of the cluster tree $T_f$ by imposing a topology on the corresponding collection of connected components $\{T_f\}$ in Definition 1. For defining a topology on $\{T_f\}$, we define the tree distance function $d_{T_f}$ in Definition 5, and impose the metric topology induced from the tree distance function. Using a distance function for topology not only eases formulating topology but also enables us to inherit all the good properties of the metric topology.

The desired tree distance function $d_{T_f} : \{T_f\} \times \{T_f\} \to [0, \infty)$ is based on the merge height function $m_f$ in Definition 2. For later use in the proof, we define the tree distance function $d_{T_f}$ on both $\mathcal{X}$ and $\{T_f\}$ as follows:

**Definition 5.** *Let $f : \mathcal{X} \to [0, \infty)$ be a function, and $T_f$ be its cluster tree in Definition 1. For any two points $x, y \in \mathcal{X}$, the tree distance function $d_{T_f} : \mathcal{X} \times \mathcal{X} \to [0, \infty)$ of $T_f$ on $\mathcal{X}$ is defined as*

$$d_{T_f}(x, y) = f(x) + f(y) - 2m_f(x, y).$$

*Similarly, for any two clusters $C_1, C_2 \in \{T_f\}$, we first define $\lambda_1 = \sup\{\lambda : C_1 \in T_f(\lambda)\}$, and $\lambda_2$ analogously. We then define the tree distance function $d_{T_f} : \{T_f\} \times \{T_f\} \to [0, \infty)$ of $T_f$ on $\mathcal{X}$ as:*

$$d_{T_f}(C_1, C_2) = \lambda_1 + \lambda_2 - 2m_f(C_1, C_2).$$

The tree distance function $d_{T_f}$ in Definition 2 is a pseudometric on $\mathcal{X}$ and is a metric on $\{T_f\}$ as desired, proven in Lemma 4. The proof is given later in Appendix E.

**Lemma 4.** *Let $f : \mathcal{X} \to [0, \infty)$ be a function, $T_f$ be its cluster tree in Definition 1, and $d_{T_f}$ be its tree distance function in Definition 5. Then $d_{T_f}$ on $\mathcal{X}$ is a pseudometric and $d_{T_f}$ on $\{T_f\}$ is a metric.*

From the metric $d_{T_f}$ on $\{T_f\}$ in Definition 5, we impose the induced metric topology on $\{T_f\}$. We say $T_f$ is homeomorphic to $T_g$, or $T_f \cong T_g$, when their corresponding collection of connected components are homeomorphic, i.e. $\{T_f\} \cong \{T_g\}$. (Two spaces are homeomorphic if there exists a bijective continuous function between them, with a continuous inverse.)

To get some geometric understanding of the cluster tree in Definition 1, we identify edges that constitute the cluster tree. Intuitively, edges correspond to either leaves or internal branches. An edge is roughly defined as a set of clusters whose inclusion relationship with respect to clusters outside an edge are equivalent, so that when the collection of connected components is divided into edges, we observe the same inclusion relationship between representative clusters whenever any cluster is selected as a representative for each edge.

For formally defining edges, we define an interval in the cluster tree and the equivalence relation in the cluster tree. For any two clusters $A, B \in \{T_f\}$, the interval $[A, B] \subset \{T_f\}$ is defined as a set clusters that contain $A$ and are contained in $B$, i.e.

$$[A, B] := \{C \in \{T_f\} : A \subset C \subset B\},$$

The equivalence relation $\sim$ is defined as $A \sim B$ if and only if their inclusion relationship with respect to clusters outside $[A, B]$ and $[B, A]$, i.e.

$A \sim B$ if and only if

for all $C \in \{T_f\}$ such that $C \notin [A, B] \cup [B, A]$, $C \subset A$ iff $C \subset B$ and $A \subset C$ iff $B \subset C$.

Then it is easy to see that the relation $\sim$ is reflexive($A \sim A$), symmetric($A \sim B$ implies $B \sim A$), and transitive ($A \sim B$ and $B \sim C$ implies $A \sim C$). Hence the relation $\sim$ is indeed an equivalence relation, and we can consider the set of equivalence classes $\{T_f\}/_\sim$. We define the edge set $E(T_f)$ as $E(T_f) := \{T_f\}/_\sim$.

For later use, we define the partial order on the edge set $E(T_f)$) as follows: $[C_1] \leq [C_2]$ if and only if for all $A \in [C_1]$ and $B \in [C_2]$, $A \subset B$. We say that a tree $T_f$ is finite if its edge $E(T_f)$ is a finite set.

## B  The Partial Order

As discussed in Section 2, to see that the partial order $\preceq$ in Definition 4 is indeed a partial order, we need to check the reflexivity, the transitivity, and the antisymmetry. The reflexivity and the transitivity are easier to check, but to show antisymmetric, we need to show that if two trees $T_f$ and $T_g$ satisfies $T_f \preceq T_g$ and $T_g \preceq T_f$, then $T_f$ and $T_g$ are equivalent in some sense. And we give the equivalence relation as the topology on the cluster tree defined in Appendix A. The argument is formally stated in Lemma 5. The proof is done later in Appendix E.

**Lemma 5.** *Let $f, g : \mathcal{X} \to [0, \infty)$ be functions, and $T_f, T_g$ be their cluster trees in Definition 1. Then if $f, g$ are continuous and $T_f, T_g$ are finite, $T_f \preceq T_g$ and $T_g \preceq T_f$ implies that there exists a homeomorphism $\Phi : \{T_f\} \to \{T_g\}$ that preserves the root, i.e. $\Phi(\mathcal{X}) = \mathcal{X}$. Conversely, if there exists a homeomorphism $\Phi : \{T_f\} \to \{T_g\}$ that preserves the root, $T_f \preceq T_g$ and $T_g \preceq T_f$ hold.*

The partial order $\preceq$ in Definition 4 gives a formal definition of simplicity of trees, and it is used to justify pruning schemes in Section 4.2. Hence it is important to match the partial order $\preceq$ with the intuitive notions of the complexity of the tree. We provided three arguments in Section 2: (1) if $T_f \preceq T_g$ holds then it must be the case that (number of edges of $T_f$) $\leq$ (number of edges of $T_g$), (2) if $T_g$ can be obtained from $T_f$ by adding edges, then $T_f \preceq T_g$ holds, and (3) the existence of a topology preserving embedding from $\{T_f\}$ to $\{T_g\}$ implies the relationship $T_f \preceq T_g$. We formally state each item in Lemma 6, 7, and 8. Proofs of these lemmas are done later in Appendix E.

**Lemma 6.** *Let $f, g : \mathcal{X} \to [0, \infty)$ be functions, and $T_f, T_g$ be their cluster trees in Definition 1. Suppose $T_f \preceq T_g$ via $\Phi : \{T_f\} \to \{T_g\}$. Define $\bar{\Phi} : E(T_f) \to E(T_g)$ by for $[C] \in E(T_f)$ choosing any $C \in [C]$ and defining as $\bar{\Phi}([C]) = [\Phi(C)]$. Then $\bar{\Phi}$ is injective, and as a consequence, $|E(T_f)| \leq |E(T_g)|$.*

**Lemma 7.** *Let $f, g : \mathcal{X} \to [0, \infty)$ be functions, and $T_f, T_g$ be their cluster trees in Definition 1. If $T_g$ can be obtained from $T_f$ by adding edges, then $T_f \preceq T_g$ holds.*

**Lemma 8.** *Let $f, g : \mathcal{X} \to [0, \infty)$ be functions, and $T_f, T_g$ be their cluster trees in Definition 1. If there exists a one-to-one map $\Phi : \{T_f\} \to \{T_g\}$ that is a homeomorphism between $\{T_f\}$ and $\Phi(\{T_f\})$ and preserves the root, i.e. $\Phi(\mathcal{X}) = \mathcal{X}$, then $T_f \preceq T_g$ holds.*

## C  Hadamard Differentiability

**Definition 6 (see page 281 of [24]).** *Let $\mathbb{D}$ and $\mathbb{E}$ be normed spaces and let $\phi : \mathbb{D}_\phi \to \mathbb{E}$ be a map defined on a subset $\mathbb{D}_\phi \subset \mathbb{D}$. Then $\phi$ is Hadamard differentiable at $\theta$ if there exists a continuous, linear map $\phi'_\theta : \mathbb{D} \to \mathbb{E}$ such that*

$$\left\| \frac{\phi(\theta + t q_t) - \phi(\theta)}{t} - \phi'_\theta(h) \right\|_{\mathbb{E}} \to 0$$

*as $t \to 0$, for every $q_t \to q$.*

Hadamard differentiability is a key property for bootstrap inference since it is a sufficient condition for the delta method; for more details, see section 3.1 of [24]. Recall that $d_{\mathrm{MM}}$ is based on the function $d_{T_p}(x, y) = p(x) + p(y) - 2m_p(x, y)$. The following theorem shows that the function $d_{T_p}$ is not Hadamard differentiable for some pairs $(x, y)$. In our case $\mathbb{D}$ is the set of continuous functions on the sample space, $\mathbb{E}$ is the real line, $\theta = p$, $\phi(p)$ is $d_{T_p}(x, y)$ and the norm on $\mathbb{E}$ is the usual Euclidean norm.

**Theorem 9.** *Let $B(x)$ be the smallest set $B \in T_p$ such that $x \in B$. $d_{T_p}(x, y)$ is not Hadamard differentiable for $x \neq y$ when one of the following two scenarios occurs:*

*(i) $\min\{p(x), p(y)\} = p(c)$ for some critical point $c$.*

*(ii) $B(x) = B(y)$ and $p(x) = p(y)$.*

The merge distortion metric $d_{\mathrm{M}}$ is also not Hadamard differentiable.

# D Confidence Sets Constructions

## D.1 Regularity conditions on the kernel

To apply the results in [8] which imply that the bootstrap confidence set is consistent, we consider the following two assumptions.

**(K1)** The kernel function $K$ has the bounded second derivative and is symmetric, non-negative, and

$$\int x^2 K(x)dx < \infty, \qquad \int K(x)^2 dx < \infty.$$

**(K2)** The kernel function $K$ satisfies

$$\mathcal{K} = \left\{ y \mapsto K\left(\frac{x-y}{h}\right) : x \in \mathbb{R}^d, h > 0 \right\}. \tag{2}$$

We require that $\mathcal{K}$ satisfies

$$\sup_P N\left(\mathcal{K}, L_2(P), \epsilon \|F\|_{L_2(P)}\right) \leq \left(\frac{A}{\epsilon}\right)^v \tag{3}$$

for some positive numbers $A$ and $v$, where $N(T, d, \epsilon)$ denotes the $\epsilon$-covering number of the metric space $(T, d)$, $F$ is the envelope function of $\mathcal{K}$, and the supremum is taken over the whole $\mathbb{R}^d$. The $A$ and $v$ are usually called the VC characteristics of $\mathcal{K}$. The norm $\|F\|_{L_2(P)}^2 = \int |F(x)|^2 dP(x)$.

Assumption (K1) is to ensure that the variance of the KDE is bounded and $p_h$ has the bounded second derivative. This assumption is very common in statistical literature, see e.g. [22, 19]. Assumption (K2) is to regularize the complexity of the kernel function so that the supremum norm for kernel functions and their derivatives can be bounded in probability. A similar assumption appears in [11] and [14]. The Gaussian kernel and most compactly supported kernels satisfy both assumptions.

## D.2 Pruning

The goal of this section is to formally define the pruning scheme in Section 4.2. Note that when pruning leaves and internal branches, when the cumulative length is computed for each leaf and internal branch, then the pruning process can be done at once. We provide two pruning schemes in Section 4.2 in a unifying framework by defining an appropriate notion of lifetime for each edge, and deleting all insignificant edges with small lifetimes. To follow the pruning schemes in Section 4.2, we require that the lifetime of a child edge is shorter than the lifetime of a parent edge, so that we can delete edges from the top. We evaluate the lifetime of each edge by an appropriate nonnegative (possibly infinite) function life. We formally define the pruned tree $Pruned_{\text{life}, \hat{t}_\alpha}(\widehat{T}_h)$ as follows:

**Definition 7.** *Suppose the function* life $: E(\widehat{T}_h) \to [0, +\infty]$ *satisfies that* $[C_1] \leq [C_2] \implies$ life$([C_1]) \subset$ life$([C_2])$. *We define the pruned tree* $Pruned_{\text{life}, \hat{t}_\alpha}(\widehat{T}_h) : \mathbb{R} \to 2^{\mathcal{X}}$ *as*

$$Pruned_{\text{life}, \hat{t}_\alpha}(\widehat{T}_h)(\lambda) = \left\{ C \in \widehat{T}_h(\lambda - \hat{t}_\alpha) : \text{life}([C]) > \hat{t}_\alpha \right\}.$$

We suggest two life functions corresponding to two pruning schemes in Section 4.2. We first need several definitions. For any $[C] \in E(\widehat{T}_h)$, define its level as

$$level([C]) := \left\{ \lambda : \text{there exists } A \in [C] \cap \widehat{T}_h(\lambda) \right\},$$

and define its cumulative level as

$$cumlevel([C]) := \left\{ \lambda : \text{there exists } A \in \widehat{T}_h(\lambda), B \in [C] \text{ such that } A \subset B \right\}.$$

Then life$^{leaf}$ corresponds to first pruning scheme in Section 4.2, which is to prune out only insignificant leaves.

$$\text{life}^{leaf}([C]) = \begin{cases} \sup\{level([C])\} - \inf\{level([C])\} & \text{if } \inf\{level([C])\} \neq \inf\{cumlevel([C])\} \\ +\infty & \text{otherwise.} \end{cases}$$

And $\text{life}^{top}$ corresponds to second pruning scheme in Section 4.2, which is to prune out insignificant edges from the top.

$$\text{life}^{top}([C]) = \sup\{cumlevel([C])\} - \inf\{cumlevel([C])\}.$$

Note that $\text{life}^{leaf}$ is lower bounded by $\text{life}^{top}$. In fact, for any life function that is lower bounded by $\text{life}^{top}$, the pruned tree $Pruned_{\text{life},\hat{t}_\alpha}$ is a valid tree in the confidence set $\widehat{C}_\alpha$ that is simpler than the original estimate $\widehat{T}_h$, so that the pruned tree is the desired tree as discussed in Section 4.2. We formally state as follows. The proof is given in Appendix G

**Lemma 10.** *Suppose that the* life *function satisfies: for all* $[C] \in E(\widehat{T}_h)$, $\text{life}^{top}([C]) \leq \text{life}([C])$. *Then*

(i) $Pruned_{\text{life},\hat{t}_\alpha}(\widehat{T}_h) \preceq T_{\widehat{p}_h}$.

(ii) *there exists a function* $\widetilde{p}$ *such that* $T_{\widetilde{p}} = Pruned_{\text{life},\hat{t}_\alpha}(\widehat{T}_h)$.

(iii) $\widetilde{p}$ *in (ii) satisfies* $\widetilde{p} \in \widehat{C}_\alpha$.

**Remark:** It can be shown that complete pruning — simultaneously removing all leaves and branches with length less than $2\hat{t}_\alpha$ — can in general yield a tree that is outside the confidence set. For example, see Figure 3. If we do complete pruning to this tree, we will get the trivial tree.

## E  Proofs for Appendix A and B

### E.1  Proof of Lemma 4

**Lemma 4.** *Let* $f : \mathcal{X} \to [0, \infty)$ *be a function,* $T_f$ *be its cluster tree in Definition 1, and* $d_{T_f}$ *be its tree distance function in Definition 5. Then* $d_{T_f}$ *on* $\mathcal{X}$ *is a pseudometric and* $d_{T_f}$ *on* $\{T_f\}$ *is a metric.*

**Proof.**

First, we show that $d_{T_f}$ on $\mathcal{X}$ is a pseudometric. To do this, we need to show non-negativity($d_{T_f}(x,y) \geq 0$), $x = y$ implying $d_{T_f}(x,y) = 0$, symmetry($d_{T_f}(x,y) = d_{T_f}(y,x)$), and subadditivity($d_{T_f}(x,y) + d_{T_f}(y,z) \leq d_{T_f}(x,z)$).

For non-negativity, note that for all $x, y \in \mathcal{X}$, $m_f(x,y) \leq \min\{f(x), f(y)\}$, so

$$d_{T_f}(x,y) = f(x) + f(y) - 2m_f(x,y) \geq 0. \tag{4}$$

For $x = y$ implying $d_{T_f}(x,y) = 0$, $x = y$ implies $m_f(x,y) = f(x) = f(y)$, so

$$x = y \implies d_{T_f}(x,y) = 0. \tag{5}$$

For symmetry, since $m_f(x,y) = m_f(y,x)$,

$$d_{T_f}(x,y) = d_{T_f}(y,x). \tag{6}$$

For subadditivity, note first that $m_f(x,y) \leq f(y)$ and $m_f(y,z) \leq f(y)$ holds, so

$$\max\{m_f(x,y), m_f(y,z)\} \leq f(y). \tag{7}$$

And also note that there exists $C_{xy}, C_{yz} \in T_f(\min\{m_f(x,y), m_f(y,z)\})$ that satisfies $x, y \in C_{xy}$ and $y, z \subset C_{yz}$. Then $y \in C_{xy} \cap C_{yz} \neq \emptyset$, so $x, z \in C_{xy} = C_{yz}$. Then from definition of $m_f(x,z)$, this implies that

$$\min\{m_f(x,y), m_f(y,z)\} \leq m_f(x,z). \tag{8}$$

And by applying (7) and (8), $d_{T_f}(x,y) + d_{T_f}(y,z)$ is upper bounded by $d_{T_f}(x,z)$ as

$$\begin{aligned}
& d_{T_f}(x,y) + d_{T_f}(y,z) \\
&= f(x) + f(y) - 2m_f(x,y) + f(y) + f(z) - 2m_f(y,z) \\
&= f(x) + f(z) - 2\left(\min\{m_f(x,y), m_f(y,z)\} + \max\{m_f(x,y), m_f(y,z)\} - f(y)\right) \\
&\geq f(x) + f(z) - 2m_f(x,z) \\
&= d_{T_f}(x,z).
\end{aligned} \tag{9}$$

Hence (4), (5), (6), and (9) implies that $d_{T_f}$ on $\mathcal{X}$ is a pseudometric.

Second, we show that $d_{T_f}$ on $\{T_f\}$ is a metric. To do this, we need to show non-negativity($d_{T_f}(x, y) \geq 0$), identity of indiscernibles($x = y \iff d_{T_f}(x, y) = 0$), symmetry($d_{T_f}(x, y) = d_{T_f}(y, x)$), and subadditivity($d_{T_f}(x, y) + d_{T_f}(y, z) \leq d_{T_f}(x, z)$).

For nonnegativity, note that if $C_1 \in T_f(\lambda_1)$ and $C_2 \in T_f(\lambda_2)$, then $m_f(C_1, C_2) \leq \min\{\lambda_1, \lambda_2\}$, so

$$d_{T_f}(C_1, C_2) = \lambda_1 + \lambda_2 - 2m_f(C_1, C_2) \geq 0. \tag{10}$$

For identity of indiscernibles, $C_1 = C_2$ implies $m_f(C_1, C_2) = \lambda_1 = \lambda_2$, so

$$C_1 = C_2 \implies d_{T_f}(C_1, C_2) = 0. \tag{11}$$

And conversely, $d_{T_f}(C_1, C_2) = 0$ implies $\lambda_1 = \lambda_2 = m_f(C_1, C_2)$, so there exists $C \in T_f(\lambda_1)$ such that $C_1 \subset C$ and $C_2 \subset C$. Then since $C_1, C_2, C \in T_f(\lambda_1)$, so $C_1 \cap C \neq \emptyset$ implies $C_1 = C$ and similarly $C_2 = C$, so

$$d_{T_f}(C_1, C_2) = 0 \implies C_1 = C_2. \tag{12}$$

Hence (11) and (12) implies identity of indiscernibles as

$$C_1 = C_2 \iff d_{T_f}(C_1, C_2) = 0. \tag{13}$$

For symmetry, since $m_f(C_1, C_2) = m_f(C_2, C_1)$,

$$d_{T_f}(C_1, C_2) = d_{T_f}(C_2, C_1). \tag{14}$$

For subadditivity, note that $m_f(C_1, C_2) \leq \lambda_2$ and $m_f(C_2, C_3) \leq \lambda_2$ holds, so

$$\max\{m_f(C_1, C_2), m_f(C_2, C_3)\} \leq \lambda_2. \tag{15}$$

And also note that there exists $C_{12}, C_{23} \in T_f(\min\{m_f(C_1, C_2), m_f(C_2, C_3)\})$ that satisfies $C_1, C_2 \subset C_{12}$ and $C_2, C_3 \subset C_{23}$. Then $C_2 \subset C_{12} \cap C_{23} \neq \emptyset$, so $C_1, C_3 \in C_{12} = C_{23}$. Then from definition of $m_f(C_1, C_3)$, this implies that

$$\min\{m_f(C_1, C_2), m_f(C_2, C_3)\} \leq m_f(C_1, C_3). \tag{16}$$

And by applying (15) and (16), $d_{T_f}(C_1, C_2) + d_{T_f}(C_2, C_3)$ is upper bounded by $d_{T_f}(C_1, C_3)$ as

$$
\begin{aligned}
& d_{T_f}(C_1, C_2) + d_{T_f}(C_2, C_3) \\
&= \lambda_1 + \lambda_2 - 2m_f(C_1, C_2) + \lambda_2 + \lambda_3 - 2m_f(C_2, C_3) \\
&= \lambda_1 + \lambda_3 - 2\left(\min\{m_f(C_1, C_2), m_f(C_2, C_3)\} + \max\{m_f(C_1, C_2), m_f(C_2, C_3)\} - \lambda_2\right) \\
&\geq \lambda_1 + \lambda_3 - 2m_f(C_1, C_3) \\
&= d_{T_f}(C_1, C_3).
\end{aligned} \tag{17}
$$

Hence (10), (13), (14), and (17) $d_{T_f}$ on $\{T_f\}$ is a metric.

$\square$

### E.2 Proof of Lemma 5

**Lemma 5.** *Let $f, g : \mathcal{X} \to [0, \infty)$ be functions, and $T_f, T_g$ be their cluster trees in Definition 1. Then if $f, g$ are continuous and $T_f, T_g$ are finite, $T_f \preceq T_g$ and $T_g \preceq T_f$ implies that there exists a homeomorphism $\Phi : \{T_f\} \to \{T_g\}$ that preserves the root, i.e. $\Phi(\mathcal{X}) = \mathcal{X}$. Conversely, if there exists a homeomorphism $\Phi : \{T_f\} \to \{T_g\}$ that preserves the root, $T_f \preceq T_g$ and $T_g \preceq T_f$ hold.*

**Proof.**

First, we show that $T_f \preceq T_g$ and $T_g \preceq T_f$ implies homeomorphism. Let $\Phi : \{T_f\} \to \{T_g\}$ be the map that gives the partial order $T_f \preceq T_g$ in Definition 4. Then from Lemma 6, $\bar{\Phi} : E(T_f) \to E(T_g)$ is injective and $|E(T_f)| \leq |E(T_g)|$. With a similar argument, $|E(T_g)| \leq |E(T_f)|$ holds, so

$$|E(T_f)| = |E(T_g)|.$$

Since we assumed that $T_f$ and $T_g$ are finite, i.e. $|E(T_f)|$ and $|E(T_g)|$ are finite, $\bar{\Phi}$ becomes a bijection.

Now, let $[C_1]$ and $[C_2]$ be adjacent edges in $E(T_f)$, and without loss of generality, assume $C_1 \subset C_2$. We argue below that $\bar{\Phi}([C_1])$ and $\bar{\Phi}([C_2])$ are also adjacent edges. Then $\Phi(C_1) \subset \Phi(C_2)$ holds from

Definition 4, and since $\bar{\Phi}$ is bijective, $[\Phi(C_1)] = \bar{\Phi}([C_1])$ and $[\Phi(C_2)] = \bar{\Phi}([C_2])$ holds. Suppose there exists $\widetilde{C}_3 \in \{T_g\}$ such that $[\widetilde{C}_3] \notin \{\bar{\Phi}([C_1]), \bar{\Phi}([C_2])\}$ and $\Phi(C_1) \subset \widetilde{C}_3 \subset \Phi(C_2)$. Then since $\bar{\Phi}$ is bijective, there exists $C_3 \in \{T_f\}$ such that $[\Phi(C_3)] = [\widetilde{C}_3]$. Then $\Phi(C_1) \subset \widetilde{C}_3 \subset \Phi(C_2)$ implies that $C_1 \subset C_3 \subset C_2$, and $\bar{\Phi}$ being a bijection implies that $[C_3] \notin \{[C_1], [C_3]\}$. This is a contradiction since $[C_1]$ and $[C_2]$ are adjacent edges. Hence there is no such $\widetilde{C}_3$, and $\bar{\Phi}([C_1])$ and $\bar{\Phi}([C_2])$ are adjacent edges. Therefore, $\bar{\Phi} : E(T_f) \to E(T_g)$ is a bijective map that sends adjacent edges to adjacent edges, and also sends root edge to root edge.

Then combining $\bar{\Phi} : E(T_f) \to E(T_g)$ being bijective sending adjacent edges to adjacent edges and root edge to root edge, and $f, g$ being continuous, the map $\bar{\Phi} : E(T_f) \to E(T_g)$ can be extended to a homeomorphism $\{T_g\} \to \{T_f\}$ that preserves the root.

Second, the part that homeomorphism implies $T_f \preceq T_g$ and $T_g \preceq T_f$ follows by Lemma 8. $\square$

### E.3  Proof of Lemma 6

**Lemma 6.** *Let $f, g : \mathcal{X} \to [0, \infty)$ be functions, and $T_f, T_g$ be their cluster trees in Definition 1. Suppose $T_f \preceq T_g$ via $\Phi : \{T_f\} \to \{T_g\}$. Define $\bar{\Phi} : E(T_f) \to E(T_g)$ by for $[C] \in E(T_f)$ choosing any $C \in [C]$ and defining as $\bar{\Phi}([C]) = [\Phi(C)]$. Then $\bar{\Phi}$ is injective, and as a consequence, $|E(T_f)| \le |E(T_g)|$.*

**Proof.**

We will first show that equivalence relation on $\{T_g\}$ implies equivalence relation on $\{T_f\}$, i.e.

$$\Phi(C_1) \sim \Phi(C_2) \implies C_1 \sim C_2. \tag{18}$$

Suppose $\Phi(C_1) \sim \Phi(C_2)$ in $\{T_g\}$. Then from Definition 4 of $\Phi$, for any $C \in \{T_f\}$ such that $C \notin [C_1, C_2] \cup [C_2, C_1]$, $\Phi(C) \notin [\Phi(C_1), \Phi(C_2)] \cup [\Phi(C_2), \Phi(C_1)]$ holds. Then from definition of $\Phi(C_1) \sim \Phi(C_2)$,

$$\Phi(C) \subset \Phi(C_1) \text{ iff } \Phi(C) \subset \Phi(C_2) \text{ and } \Phi(C_1) \subset \Phi(C) \text{ iff } \Phi(C_2) \subset \Phi(C).$$

Then again from Definition 4 of $\Phi$, equivalence relation holds for $C_1$ and $C_2$ holds as well, i.e.

$$C \subset C_1 \text{ iff } C \subset C_2 \text{ and } C_1 \subset C \text{ iff } C_2 \subset C.$$

Hence (18) is shown, and this implies that

$$\begin{aligned}
\bar{\Phi}([C_1]) = \bar{\Phi}([C_2]) &\implies [\Phi(C_1)] = [\Phi(C_2)] \\
&\implies \Phi(C_1) \sim \Phi(C_2) \\
&\implies C_1 \sim C_2 \\
&\implies [C_1] = [C_2],
\end{aligned}$$

so $\bar{\Phi}$ is injective. $\square$

### E.4  Proof of Lemma 7

**Lemma 7.** *Let $f, g : \mathcal{X} \to [0, \infty)$ be functions, and $T_f, T_g$ be their cluster trees in Definition 1. If $T_g$ can be obtained from $T_f$ by adding edges, then $T_f \preceq T_g$ holds.*

**Proof.** Since $T_g$ can be obtained from $T_f$ by adding edges, there is a map $\Phi : T_f \to T_g$ which preserves order, i.e. $C_1 \subset C_2$ if and only if $\Phi(C_1) \subset \Phi(C_2)$. Hence $T_f \preceq T_g$ holds. $\square$

### E.5  Proof of Lemma 8

**Lemma 8.** *Let $f, g : \mathcal{X} \to [0, \infty)$ be functions, and $T_f, T_g$ be their cluster trees in Definition 1. If there exists a one-to-one map $\Phi : \{T_f\} \to \{T_g\}$ that is a homeomorphism between $\{T_f\}$ and $\Phi(\{T_f\})$ and preserves root, i.e. $\Phi(\mathcal{X}) = \mathcal{X}$, then $T_f \preceq T_g$ holds.*

**Proof.** For any $C \in \{T_f\}$, note that $[C, \mathcal{X}] \subset \{T_f\}$ is homeomorphic to an interval, hence $\Phi([C, \mathcal{X}]) \subset \{T_g\}$ is also homeomorphic to an interval. Since $\{T_g\}$ is topologically a tree, an interval in a tree with fixed boundary points is uniquely determined, i.e.

$$\Phi([C, \mathcal{X}]) = [\Phi(C), \Phi(\mathcal{X})] = [\Phi(C), \mathcal{X}]. \tag{19}$$

For showing $T_f \preceq T_g$, we need to argue that for all $C_1, C_2 \in \{T_f\}$, $C_1 \subset C_2$ holds if and only if $\Phi(C_1) \subset \Phi(C_2)$. For only if direction, suppose $C_1 \subset C_2$. Then $C_2 \in [C_1, \mathcal{X}]$, so Definition 4 and (19) implies

$$\Phi(C_2) \subset \Phi([C_1, \mathcal{X}]) = [\Phi(C_1), \mathcal{X}].$$

And this implies

$$\Phi(C_1) \subset \Phi(C_2). \tag{20}$$

For if direction, suppose $\Phi(C_1) \subset \Phi(C_2)$. Then since $\Phi^{-1} : \Phi(\{T_f\}) \to \{T_f\}$ is also an homeomorphism with $\Phi^{-1}(\mathcal{X}) = \mathcal{X}$, hence by repeating above argument, we have

$$C_1 = \Phi^{-1}(\Phi(C_1)) \subset \Phi^{-1}(\Phi(C_2)) = C_2. \tag{21}$$

Hence (20) and (21) implies $T_f \preceq T_g$. $\square$

# F   Proofs for Section 3 and Appendix C

## F.1   Proof of Lemma 1 and extreme cases

**Lemma 1.** *For any densities $p$ and $q$, the following relationships hold:*

- (i) *When $p$ and $q$ are continuous, then $d_\infty(T_p, T_q) = d_M(T_p, T_q)$.*
- (ii) $d_{MM}(T_p, T_q) \leq 4d_\infty(T_p, T_q)$.
- (iii) $d_{MM}(T_p, T_q) \geq d_\infty(T_p, T_q) - a$, *where $a$ is defined as above. Additionally when $\mu(\mathcal{X}) = \infty$, then $d_{MM}(T_p, T_q) \geq d_\infty(T_p, T_q)$.*

**Proof.**

(i)

First, we show $d_M(T_p, T_q) \leq d_\infty(T_p, T_q)$. Note that this part is implicitly shown in Eldridge et al. [12, Proof of Theorem 6]. For all $\epsilon > 0$ and for any $x, y \in \mathcal{X}$, let $C_0 \in T_p(m_p(x, y) - \epsilon)$ with $x, y \in C_0$. Then for all $z \in C_0$, $q(z)$ is lower bounded as

$$q(z) > p(z) - d_\infty(T_p, T_q)$$
$$\geq m_p(x, y) - \epsilon - d_\infty(T_p, T_q),$$

so $C_0 \subset q^{-1}(m_p(x, y) - \epsilon - d_\infty(T_p, T_q), \infty)$ and $C_0$ is connected, so $x$ and $y$ are in the same connected component of $q^{-1}(m_p(x, y) - \epsilon - d_\infty(T_p, T_q), \infty)$, which implies

$$m_q(x, y) \leq m_p(x, y) - \epsilon - d_\infty(T_p, T_q). \tag{22}$$

A similar argument holds for other direction as

$$m_p(x, y) \leq m_q(x, y) - \epsilon - d_\infty(T_p, T_q), \tag{23}$$

so (22) and (23) being held for all $\epsilon > 0$ implies

$$|m_p(x, y) - m_q(x, y)| \leq d_\infty(T_p, T_q). \tag{24}$$

And taking sup over all $x, y \in \mathcal{X}$ in (24) $d_M(T_p, T_q)$ is upper bounded by $d_\infty(T_p, T_q)$, i.e.

$$d_M(T_p, T_q) \leq d_\infty(T_p, T_q). \tag{25}$$

Second, we show $d_M(T_p, T_q) \geq d_\infty(T_p, T_q)$. For all $\epsilon > 0$, Let $x$ be such that $|p(x) - q(x)| > d_\infty(T_p, T_q) - \frac{\epsilon}{2}$. Then since $p$ and $q$ are continuous, there exists $\delta > 0$ such that

$$B(x, \delta) \subset p^{-1}\left(p(x) - \frac{\epsilon}{2}, \infty\right) \cap q^{-1}\left(q(x) - \frac{\epsilon}{2}, \infty\right).$$

Then for any $y \in B(x, \delta)$, since $B(x, \delta)$ is connected, $p(x) - \frac{\epsilon}{2} \leq m_p(x, y) \leq p(x)$ holds and $q(x) - \frac{\epsilon}{2} \leq m_q(x, y) \leq q(x)$, so

$$|m_p(x, y) - m_q(x, y)| \geq |p(x) - q(x)| - \frac{\epsilon}{2}$$
$$> d_\infty(T_p, T_q) - \epsilon.$$

Since this holds for any $\epsilon > 0$, $d_M(T_p, T_q)$ is lower bounded by $d_\infty(T_p, T_q)$, i.e.

$$d_M(T_p, T_q) \geq d_\infty(T_p, T_q). \tag{26}$$

(25) and (26) implies $d_\infty(T_p, T_q) = d_M(T_p, T_q)$.

(ii)

We have already seen that for all $x, y \in \mathcal{X}$, $|m_p(x, y) - m_q(x, y)| \leq d_\infty(T_p, T_q)$ in (24). Hence for all $x, y \in \mathcal{X}$,

$$
\begin{aligned}
&|[p(x) + p(y) - 2m_p(x, y)] - [q(x) + q(y) - 2m_q(x, y)]| \\
&\leq |p(x) - q(x)| + |p(y) - q(y)| + 2|m_p(x, y) - m_q(x, y)| \\
&\leq 4d_\infty(T_p, T_q).
\end{aligned}
$$

Since this holds for all $x, y \in \mathcal{X}$, so

$$d_{\mathrm{MM}}(T_p, T_q) \leq 4d_\infty(T_p, T_q).$$

(iii)

For all $\epsilon > 0$, Let $x$ be such that $|p(x) - q(x)| > d_\infty(T_p, T_q) - \frac{\epsilon}{2}$, and without loss of generality assume that $p(x) > q(x)$. Let $y$ be such that $p(y) + q(y) < \inf_x(p(x) + q(x)) + \frac{\epsilon}{2}$. Then $m_p(x, y) \leq p(y)$ holds, and since $\mathcal{X}$ is connected, $q_{\inf} \leq m_q(x, y)$ holds. Hence

$$
\begin{aligned}
&[p(x) + p(y) - 2m_p(x, y)] - [q(x) + q(y) - 2m_q(x, y)] \\
&\geq [p(x) + p(y) - 2p(y)] - [q(x) + q(y) - 2q_{\inf}] \\
&= p(x) - q(x) - (p(y) + q(y) - 2q_{\inf}) \\
&> d_\infty(T_p, T_q) - \left(\inf_x(p(x) + q(x)) - 2q_{\inf}\right) - \epsilon \\
&\geq d_\infty(T_p, T_q) - a - \epsilon,
\end{aligned}
$$

where $a = \inf_{x \in \mathcal{X}}(p(x) + q(x)) - 2\min\{p_{\inf}, q_{\inf}\}$. Since this holds for all $\epsilon > 0$, we have

$$d_{\mathrm{MM}}(T_p, T_q) \geq d_\infty(T_p, T_q) - a.$$

□

Hence $0 \leq d_{\mathrm{MM}}(T_p, T_q) \leq 4d_\infty(T_p, T_q)$ holds. And both extreme cases can happen, i.e. $d_{\mathrm{MM}}(T_p, T_q) = 4d_\infty(T_p, T_q) > 0$ and $d_{\mathrm{MM}}(T_p, T_q) = 0$, $d_\infty(T_p, T_q) > 0$ can happens.

**Lemma 11.** *There exists densities* $p, q$ *for both* $d_{\mathrm{MM}}(T_p, T_q) = 4d_\infty(T_p, T_q) > 0$ *and* $d_{\mathrm{MM}}(T_p, T_q) = 0$, $d_\infty(T_p, T_q) > 0$.

**Proof.** Let $\mathcal{X} = \mathbb{R}$, $p(x) = I(x \in [0, 1])$ and $q(x) = 2I\left(x \in \left[0, \frac{1}{4}\right]\right) + 2I\left(x \in \left[\frac{3}{4}, 1\right]\right)$. Then $d_\infty(T_p, T_q) = 1$. And with $x = \frac{1}{8}$ and $y = \frac{7}{8}$,

$$
\begin{aligned}
|[p(x) + p(y) - 2m_p(x, y)] - [q(x) + q(y) - 2m_q(x, y)]| &= |[1 + 1 - 2] - [2 + 2 - 0]| \\
&= 4,
\end{aligned}
$$

hence $d_{\mathrm{MM}}(T_p, T_q) = 4d_\infty(T_p, T_q)$.

Let $\mathcal{X} = [0, 1)$, $p(x) = 2I\left(x \in \left[0, \frac{1}{2}\right)\right)$ and $q(x) = 2I\left(x \in \left[\frac{1}{2}, 1\right)\right)$. Then $d_\infty(T_p, T_q) = 2$. And for any $x \in \left[0, \frac{1}{2}\right)$ and $y \in \left[\frac{1}{2}, 1\right)$,

$$
\begin{aligned}
|[p(x) + p(y) - 2m_p(x, y)] - [q(x) + q(y) - 2m_q(x, y)]| &= |(2 + 0 - 0) + (0 + 2 - 0)| \\
&= 0.
\end{aligned}
$$

A similar case holds for $x \in \left[\frac{1}{2}, 1\right)$ and $y \in \left[0, \frac{1}{2}\right)$. And for any $x, y \in \left[0, \frac{1}{2}\right)$,

$$
\begin{aligned}
|[p(x) + p(y) - 2m_p(x, y)] - [q(x) + q(y) - 2m_q(x, y)]| &= |(2 + 2 - 4) + (0 + 0 - 0)| \\
&= 0.
\end{aligned}
$$

and a similar case holds for $x, y \in \left[\frac{1}{2}, 1\right)$. Hence $d_{\mathrm{MM}}(T_p, T_q) = 0$. □

Figure 6: The example used in the proof of Theorem 9.

### F.2   Proof of Theorem 9

**Theorem 9.**  *Let $B(x)$ be the smallest set $B \in T_p$ such that $x \in B$. $d_{T_p}(x,y)$ is not Hadamard differentiable for $x \neq y$ when one of the following two scenarios occurs:*

*(i) $\min\{p(x), p(y)\} = p(c)$ for some critical point $c$.*

*(ii) $B(x) = B(y)$ and $p(x) = p(y)$.*

**Proof.** For $x, y \in \mathbb{K}$, note that the merge height satisfies

$$m_p(x,y) = \min\{t : (x,y) \text{ are in the same connected component of } L(t)\}.$$

Recall that

$$d_{T_p}(x,y) = p(x) + p(y) - 2m_p(x,y).$$

Note that the modified merge distortion metric is $d_{\mathrm{MM}}(p,q) = \sup_{x,y} |d_{T_p}(x,y) - d_{T_q}(x,y)|$.

A feature of the merge height is that

$$m_p(x,y) = p(x) \Rightarrow B(y) \subset B(x)$$
$$m_p(x,y) = p(y) \Rightarrow B(x) \subset B(y)$$
$$m_p(x,y) \neq p(y) \text{ or } p(x) \Rightarrow \exists c(x,y) \in \mathcal{C} \ \ s.t. \ \ m_p(x,y) = p(c(x,y)).$$

where $\mathcal{C}$ is the collection of all critical points. Thus, we have

$$d_{T_p}(x,y) = \begin{cases} p(x) - p(y) & \text{if } B(y) \subset B(x) \\ p(y) - p(x) & \text{if } B(x) \subset B(y) \\ p(x) + p(y) - 2p(c(x,y)) & \text{otherwise} \end{cases}.$$

**Case 1:**
We pick a pair of $x_0, y_0$ as in Figure 6. Now we consider a smooth symmetric function $g(x) > 0$ such that it peaks at 0 and monotonically decay and has support $[-\delta, \delta]$ for some small $\delta > 0$. We pick $\delta$ small enough such that $p_\epsilon(x_0) = p(x_0), p_\epsilon(y_0) = p(y_0)$. For simplicity, let $g(0) = \max_x g(x) = 1$.

Now consider perturbing $p(x)$ along $g(x - c)$ with amount $\epsilon$. Namely, we define

$$p_\epsilon(x) = p(x) + \epsilon \cdot g(x - c).$$

For notational convenience, define $\xi_{p,\epsilon} = d_{T_{p_\epsilon}}(x_0, y_0)$. When $|\epsilon|$ is sufficiently small, define

$$\xi_{p,\epsilon}(x_0, y_0) = d_{T_p}(x_0, y_0) \quad \text{if } \epsilon > 0,$$
$$\xi_{p,\epsilon}(x_0, y_0) = d_{T_p}(x_0, y_0) - 2\epsilon \quad \text{if } \epsilon < 0.$$

This is because when $\epsilon > 0$, the $p_\epsilon(c) > p(c)$, so the merge height for $x_0, y_0$ using $p_\epsilon$ is still the same as $p(y_0)$, which implies $\xi_{p,\epsilon}(x_0, y_0) = d_{T_p}(x_0, y_0)$. On the other hand, when $\epsilon < 0$, $p_\epsilon(c) < p(c)$, so the merge height is no longer $p(y_0)$ but $p_\epsilon(c)$. Then using the fact that $|\epsilon| = p(c) - p_\epsilon(c)$ we obtain the result.

Now we show that $d_{T_p}(x_0, y_0)$ is not Hadamard differentiable. In this case, $\phi(p) = \xi_p(x_0, y_0)$. First, we pick a sequence of $\epsilon_n$ such that $\epsilon_n \to 0$ and $\epsilon_n > 0$ if $n$ is even and $\epsilon_n < 0$ if $n$ is odd. Plugging $t \equiv \epsilon_n$ and $q_t = g$ into the definition of Hadamard differentiability, we have

$$\phi'(p) \equiv \frac{\xi_{p,\epsilon_n}(x_0, y_0) - d_{T_p}(x_0, y_0)}{\epsilon_n}$$

is alternating between 0 and 2, so it does not converge. This shows that the function $d_{T_p}(x, y)$ at such a pair of $(x_0, y_0)$ is non-Hadamard differentiable.

**Case 2:**
The proof of this case uses the similar idea as the proof of case 1. We pick the pair $(x_0, y_0)$ satisfying the desire conditions. We consider the same function $g$ but now we perturb $p$ by

$$p_\epsilon(x) = p(x) + \epsilon \cdot g(x - x_0),$$

and as long as $\delta$ is small, we will have $p_\epsilon(y_0) = p(y_0)$. Since $B(x_0) = B(y_0)$ and $p(x_0) = p(y_0)$, $d_{T_p}(x_0, y_0) = 0$. When $\epsilon > 0$, $\xi_{p,\epsilon}(x_0, y_0) = \epsilon$, and on the other hand, when $\epsilon < 0$, $\delta_\epsilon(x_0, y_0) = -\epsilon$.

In this case, again, $\phi(p) = \xi_p(x_0, y_0)$. Now we use the similar trick as case 1: picking a sequence of $\epsilon_n$ such that $\epsilon_n \to 0$ and $\epsilon_n > 0$ if $n$ is even and $\epsilon_n < 0$ if $n$ is odd. Under this sequence of $\epsilon_n$, the 'derivative' along $g$

$$\phi'(p) \equiv \frac{\xi_{p,\epsilon_n}(x_0, y_0) - d_{T_p}(x_0, y_0)}{\epsilon_n}$$

is alternating between 1 and $-1$, so it does not converge. Thus, $d_{T_p}(x, y)$ at such a pair of $(x_0, y_0)$ is non-Hadamard differentiable. $\square$

# G   Proofs for Section 4 and Appendix D

## G.1   Proof of Lemma 2

**Lemma 2.** *Let $p_h = \mathbb{E}[\widehat{p}_h]$ where $\widehat{p}_h$ is the kernel estimator with bandwidth $h$. We assume that $p$ is a Morse function supported on a compact set with finitely many, distinct, critical values. There exists $h_0 > 0$ such that for all $0 < h < h_0$, $T_p$ and $T_{p_h}$ have the same topology in Appendix A.*

**Proof.** Let $S$ be the compact support of $p$. By the classical stability properties of the Morse function, there exists a constant $C_0 > 0$ such that for any other smooth function $q : S \to \mathbb{R}$ with $\|q - p\|_\infty, \|\nabla q - \nabla p\|_\infty, \|\nabla^2 q - \nabla^2 p\|_\infty < C_0$, $q$ is a Morse function. Moreover, there exist two diffeomorphisms $h : \mathbb{R} \to \mathbb{R}$ and $\phi : S \to S$ such that $q = h \circ p \circ \phi$ See e.g., proof of [6, Lemma 16]. Further, $h$ should be nondecreasing if $C_0$ is small enough. Hence for any $C \in T_p(\lambda)$, since $q \circ \phi^{-1}(C) = h \circ p(C)$, so $\phi^{-1}(C)$ is a connected component of $T_q(h(\lambda))$. Now define $\Phi : \{T_p\} \to \{T_q\}$ as $\Phi(C) = \phi^{-1}(C)$. Then since $\phi$ is a diffeomorphism, $C_1 \subset C_2$ if and only if $\Phi(C_1) = \phi^{-1}(C_1) \subset \phi^{-1}(C_2) = \Phi(C_2)$, hence $T_p \preceq T_q$ holds. And from $p \circ \phi = h^{-1} \circ q$, we can similarly show $T_q \preceq T_p$ as well. Hence from Lemma 5, two trees $T_p$ and $T_q$ are topologically equivalent according to the topology in Appendix A.

Now by the nonparametric theory (see e.g. page 144-145 of [19], and [22]), there is a constant $C_1 > 0$ such that $\|p_h - p\|_{2,\max} \leq C_1 h^2$ when $h < 1$. Thus, when $0 \leq h \leq \sqrt{\frac{C_0}{C_1}}$, $T_h = T_{p_h}$ and $T = T_p$ have the same topology. $\square$

## G.2   Proof of Lemma 10

**Lemma 10.** *Suppose that the* life *function satisfies: for all $[C] \in E(\widehat{T}_h)$, $\mathrm{life}^{top}([C]) \leq \mathrm{life}([C])$. Then*

(i) $Pruned_{\mathrm{life}, \widehat{t}_\alpha}(\widehat{T}_h) \preceq T_{\widehat{p}_h}$.

(ii) *there exists a function $\widetilde{p}$ such that $T_{\widetilde{p}} = Pruned_{\mathrm{life}, \widehat{t}_\alpha}(\widehat{T}_h)$.*

(iii) $\widetilde{p}$ *in (ii) satisfies $\widetilde{p} \in \widehat{C}_\alpha$.*

**Proof.**

(i)

This is implied by Lemma 7.

(ii)

Note that $Pruned_{\text{life}, \widehat{t}_\alpha}(\widehat{T}_h)$ is generated by function $\widetilde{p}$ defined as

$$\widetilde{p}(x) = \sup\left\{\lambda : \text{ there exists } C \in \widehat{T}_h(\lambda) \text{ such that } x \in C \text{ and } \text{life}([C]) > 2\widehat{t}_\alpha\right\} + \widehat{t}_\alpha.$$

(iii)

Let $C_0 := \bigcup\{C : \text{life}([C]) \le 2\widehat{t}_\alpha\}$. Then note that

$$\widehat{p}(x) = \sup\left\{\lambda : \text{ there exists } C \in \widehat{T}_h(\lambda) \text{ such that } x \in C\right\},$$

so for all $x$, $\widetilde{p}(x) \le \widehat{p}(x) + \widehat{t}_\alpha$, and if $x \notin C_0$, $\widetilde{p}(x) = \widehat{p}(x) + \widehat{t}_\alpha$. Then note that

$$\left\{\lambda : \text{ there exists } C \in \widehat{T}_h(\lambda) \text{ such that } x \in C\right\}$$
$$\setminus \left\{\lambda : \text{ there exists } C \in \widehat{T}_h(\lambda) \text{ such that } x \in C \text{ and } \text{life}([C]) > 2\widehat{t}_\alpha\right\}$$
$$\subset \left\{\lambda : \text{ there exists } C \in \widehat{T}_h(\lambda) \text{ such that } x \in C \text{ and } \text{life}([C]) \le 2\widehat{t}_\alpha\right\}$$

Let $e_x := \max\left\{e : x \in \cup e, \text{life}(e) \le 2\widehat{t}_\alpha\right\}$. Then note that $x \in C$ and $\text{life}([C]) \le 2\widehat{t}_\alpha$ implies that we can find some $B \in e_x$ such that $C \subset B$, so

$$\left\{\lambda : \text{ there exists } C \in \widehat{T}_h(\lambda) \text{ such that } x \in C \text{ and } \text{life}([C]) \le 2\widehat{t}_\alpha\right\} \subset cumlevel(e_x).$$

Hence

$$\widehat{p}(x) + \widehat{t}_\alpha - \widetilde{p}(x) \le \sup\{cumlevel(e_x)\} - \inf\{cumlevel(e_x)\}$$
$$= \text{life}^{top}(e_x)$$
$$\le \text{life}(e_x) \le 2\widehat{t}_\alpha,$$

and hence

$$\widehat{p}(x) - \widehat{t}_\alpha \le \widetilde{p}(x) \le \widehat{p}(x) + \widehat{t}_\alpha.$$

$\square$