[Reviews · NeurIPS 2016]

Reviewer 1

Summary

The authors estimate the cluster tree of the unknown density of an observed sample. The estimator is the cluster tree of a kernel density estimator. The accuracy is measured in terms of the sup-norm of the underlying density, which is shown to be equal to a natural metric on the class of clustering trees. Confidence sets are proposed, based on a bootstrap procedure. Then, a procedure is defined in order to exhibit a tree in that confidence set, which has a simpler structure than the cluster tree of the kernel density estimator, yet still shows important topological features of that estimator. Experiments results on both simulated and real data are provided.

Qualitative Assessment

1. Reading this work is pleasant, and the reader can learn many things. However, the main result is a little deceiving: although the authors write, in the introduction, that estimating the cluster tree instead of the density itself overcomes the curse of the dimension, they actually have no guarantee for the accuracy of their estimator. Actually, they do not estimate T_{p_0} but, instead, T_{p_h} for a small but arbitrary bandwidth h. In order to estimate T_{p_0} accurately, the bandwidth h should be chosen small enough, i.e., smaller than some h0, but the value of h0 is unknown (this is mentioned in line 192). Hence, the curse of dimension is actually not avoided at all. 2. In Theorem 3, which is not proven, I am surprised to find no dependency in B. 3. To my opinion, all mentions to d_{MM} should be omitted.

Confidence in this Review

2-Confident (read it all; understood it all reasonably well)


Reviewer 2

Summary

The cluster tree of a probability density function provides a hierarchical representation of its high density regions. Given an iid sample from an unknown density p_0, the paper discusses statistical inference for the cluster tree associated with p_0. Specifically, a bootstrap-based confidence set for the cluster tree (of a closely related object to p_0) is proposed. The shape of the confidence set is governed by a metric defined on cluster trees, which is amenable for the task. While the confidence set is given in some sense implicitly, the authors propose methods for obtaining "informative" representatives in the confidence set, that retain "statistically significant" features of the cluster tree.

Qualitative Assessment

I found the paper original and interesting. I am wondering what relationships exist to nonparametric inference for density functions: I assume that the task of constructing a confidence set for the cluster tree of p_0 is quite different from constructing a confidence set for p_0 itself, which justifies addressing the former problem directly instead of going through the latter (and taking {T_p: p \in C_\alpha} where C_\alpha is a confidence set for p_0). For example, the focus on p_h (rather than p_0) is justifiable as far as T_{p_h} compares with T_{p_0} but not necessarily as far as p_h compares with p_0. A few comments: 1. Since the confidence set is implicit, the main concrete contribution I see in this work, is that you have a way to prune the tree while maintaining statistical guarantees; maybe this is worth emphasizing. 2. If I wanted to test H_0: T_{p_h} = T_0, I could in principle use an arbitrary statistic instead of d_{\infty}, and could use this statistic to form a confidence set by inverting the tests (as you propose). The choice of d_{\infty} (and, in fact, of any other metric you discussed) seems a little arbitrary to me. I guess senses of "optimality" of this statistics are beyond the scope of the paper, but are there further justifications of using it?

Confidence in this Review

2-Confident (read it all; understood it all reasonably well)


Reviewer 3

Summary

The article discusses an inference problem for cluster trees arising from density estimation and proposes a method of constructing confidence sets by a bootstrap. In addition, to aid visualisation, additional pruning method is explored. The authors approach the problem by topological consideration of comparing trees, and propose an $\ell_\infty$ metric-based pruning. The main contribution seems to develop the framework for its theoretical justification, though some numerical examples are also included. Overall, the problem is interesting and important, and the careful consideration of the theory is impressive, the solution provided is reasonable and practical.

Qualitative Assessment

1. Although I appreciate its importance of theoretical development, the numerical example does not seems to demonstrate well the potential of the proposed method. In particular, since the main objective is the confidence set, I wonder if there is a way to show the richness of the confidence sets so that the readers better appreciate the visualisation by pruning. 2. Related to the first point, I originally thought that the partial ordering was considered for comparing different trees, but the way the partial ordering is defined seems to require the main split should occur at exactly the same location (Fig.2), which would never happen for two different estimates, although the variation of the branching points would be small enough to define an approximation. So the partial ordering is well defined for pruning of the given estimate, but it wouldn't help for comparing the various trees within the confidence sets. Or do I miss something here? 3. The definition of the metrics in Section 3 doesn't necessarily assume that $p$ and $q$ are the density functions representing the same underlying population (like KDE estimates from two different realisations of the population), but any functions that are defined on the same domain so that they don't have to be close at all. Is it true? 4. Similarly, for $p_{inf}$ and the definition of $a$, if we consider density functions, unless they have compact support, $p_{inf}$ and $a$ would be zero. Then in Lemma 1, "more generally" means whether it has compact support or not? 5. Considering the distance with respect to the truth, intuitively as KDE has bias proportional to the second derivatives, the main error would occur at peaks and valleys, which are directly linked to the merging height. Then the equivalence of $d_\infty$ and $d_M$ is expected. However, this doesn't seem the case when we compare two different functions, the relative difference between these functions don't have to be amplified at those landmarks, yet I feel that the same principle should be relevant. Could you give some insights into the source of my confusion? 6. In figure 4, the second example (mickey mouse data), I would expect that heights would be different for the main part and two small clusters as the probabilities would be different. Why is it not the case?

Confidence in this Review

2-Confident (read it all; understood it all reasonably well)


Reviewer 4

Summary

The paper does the following: 1. investigates a variety of metrics on cluster trees and shows the equivalence of l-infinity and merge distortion norms and show why a modified merge distortion metric won't work with their analysis and they decide to go with the l-infinity norm for their analysis. 2. constructs asymptotic confidence sets for cluster trees using a KDE estimate and bootstrapping. 3. gives a pruning procedure to simplify trees in the confidence set and introduces a partial ordering of the trees.

Qualitative Assessment

A well-written and insightful paper. The construction of confidence intervals for cluster trees seems important and in the process of doing so the authors appear to have developed a good amount of new theory.

Confidence in this Review

2-Confident (read it all; understood it all reasonably well)


Reviewer 5

Summary

This paper proposed methods to construct and summarize confidence sets for the unknown true cluster tree and illustrate the proposed methods on a variety of synthetic examples and furthermore demonstrate their utility in the analysis of a Graft-versus-Host Disease (GvHD) data set.

Qualitative Assessment

My only question about this paper is that the author does not provide any compared method which make the experiment parts weaker.

Confidence in this Review

1-Less confident (might not have understood significant parts)


Reviewer 6

Summary

The paper proposed methods to construct and summarize confidence sets for the unknown true cluster tree, and introduce a partial ordering on cluster trees which was used to prune some of the statistically insignificant features of the empirical tree.

Qualitative Assessment

Clarity: - The paper is not well organized. The presentation and discussion of the main contribution (Section 4) is very limited. Also, some of the main text can be put into the supplementary material, Section 3 for example, since it seems to be not quite related to the following construction. - The title seems to be too big for the contents provided. As mentioned in Section 1, statistical inference "allow us to quantify our uncertainty, ..., as well as to rigorously test hypotheses related to the estimated cluster structure". No such hypothesis testing discussion is presented in the paper. - It would be good to provide the labels for the x- and y-axis in Figure 4 and 5 to make them self-contained. - The "feature" used is confusing. There is no clear description on what it refers to. Innovation/Significance: - It would be better to have the result on T_(p_0) combining the results of Lemma 2 and Theorem 3. - The authors failed to deliver comparisons with the existing large-sample approximate confidence intervals, which is essential. Also, no practical situation is discussed, where these complex confidence intervals can be applied and really shine. - One of the contributions is the convergence rate independent of the dimension, but this has not be validated in the experiment section. I think partial ordering and Lemma 2 are interesting by themselves, but the whole manuscript lacks a coherent presentation and strong validations.

Confidence in this Review

2-Confident (read it all; understood it all reasonably well)